Metaheuristic optimized complex-valued dilated recurrent neural network for attack detection in internet of vehicular communications

http://orcid.org/0000-0002-6882-2233 Balaji Prasanalakshmi 1 drsanaksa@gmail.com
http://orcid.org/0000-0001-6594-8861 Cengiz Korhan 2 3
http://orcid.org/0000-0001-9378-6409 Babu Sangita 4
http://orcid.org/0009-0001-6652-3512 Alqahtani Omar 1
http://orcid.org/0000-0001-7005-6489 Akleylek Sedat 5 6
1 Department of Computer Science, King Khalid University , Alqaraa , Saudi Arabia
2 College of Information Technology, University of Fujairah , Fujairah , United Arab Emirates
3 Department of Electrical-Electronics Engineering, Istinye University , Istanbul , Turkey
4 Department of Computer Science, King Khalid University , Rijal Alma , Saudi Arabia
5 Institute of Computer Science, University of Tartu , Tartu , Estonia
6 Department of Computer Engineering, Istinye University , Istanbul , Turkey
Shah Syed Hassan
Electronic publication date: 2024 Oct 31
Publication date: 2024
Volume: 10
Electronic Location ID: e2366
Received 2024 May 7; Accepted 2024 Sep 6
Copyright: © 2024 Balaji et al.
Copyright year: 2024
Copyright holder: Balaji et al.
License: This is an open access article distributed under the terms of the Creative Commons Attribution License, which permits unrestricted use, distribution, reproduction and adaptation in any medium and for any purpose provided that it is properly attributed. For attribution, the original author(s), title, publication source (PeerJ Computer Science) and either DOI or URL of the article must be cited.
License URL: https://creativecommons.org/licenses/by/4.0/

Keywords: Internet of vehicular communication, Optimal feature extraction, Enhanced exploitation in hybrid leader-based optimization, Complex-valued dilated recurrent neural network

Funding: The Deanship of Research and Graduate Studies at King Khalid University RGP1/261/45 The Deanship of Research and Graduate Studies at King Khalid University funded this work through Small group Research Project under grant number RGP1/261/45. The funders had no role in study design, data collection and analysis, decision to publish, or preparation of the manuscript.

==============================
The Internet of Vehicles (IoV) is a specialized iteration of the Internet of Things (IoT) tailored to facilitate communication and connectivity among vehicles and their environment. It harnesses the power of advanced technologies such as cloud computing, wireless communication, and data analytics to seamlessly exchange real-time data among vehicles, road-side infrastructure, traffic management systems, and other entities. The primary objectives of this real-time data exchange include enhancing road safety, reducing traffic congestion, boosting traffic flow efficiency, and enriching the driving experience. Through the IoV, vehicles can share information about traffic conditions, weather forecasts, road hazards, and other relevant data, fostering smarter, safer, and more efficient transportation networks. Developing, implementing and maintaining sophisticated techniques for detecting attacks present significant challenges and costs, which might limit their deployment, especially in smaller settings or those with constrained resources. To overcome these drawbacks, this article outlines developing an innovative attack detection model for the IoV using advanced deep learning techniques. The model aims to enhance security in vehicular networks by efficiently identifying attacks. Initially, data is collected from online databases and subjected to an optimal feature extraction process. During this phase, the Enhanced Exploitation in Hybrid Leader-based Optimization (EEHLO) method is employed to select the optimal features. These features are utilized by a Complex-Valued Dilated Recurrent Neural Network (CV-DRNN) to detect attacks within vehicle networks accurately. The performance of this novel attack detection model is rigorously evaluated and compared with that of traditional models using a variety of metrics.

Introduction

The rapid advancements in 5G communications technology have been a cornerstone in enabling features such as autopilot, cloud service integration and the Internet of Vehicles (IoV), ensuring robust in-vehicle connectivity (Gao et al., 2019). However, the internet’s infrastructure still presents limitations. The IoV is divided into two subsets: the intra-vehicular and inter-vehicular networks. The former encompasses the network of electronic devices and sensors within a vehicle, interconnected through a central computer system for message exchange and task execution (Hoang & Kim, 2023). The latter, leveraging Vehicle-to-Everything (V2X) technology, facilitates communication between vehicles and external devices, including other vehicles, signal antennas, and road infrastructure (Jeong et al., 2021). This external communication is crucial for achieving a higher level of technological sophistication but also introduces vulnerabilities to security breaches, particularly from hackers targeting the vehicle’s electronic control units (ECUs) through cyber-attacks like denial of service and failure attacks (Luo, 2022). Various protection mechanisms have been deployed within ECUs to mitigate these security risks, with intrusion detection systems emerging as the most effective (Oseni et al., 2023). These systems are highly accurate in identifying malicious activities and offer the benefit of low computational demand compared to other security measures (Shahriar et al., 2023). The expansion of smart car connectivity is paralleled by increased security threats (Cui et al., 2023). The IoV network is susceptible to hacking, which can lead to vehicle accidents and operational instability.

An illustrative incident involved two hackers who compromised a vehicle’s steering and braking systems, executing dangerous maneuvers swiftly (Elsayed et al., 2022). Monitoring network activity and detecting malicious attempts is crucial to combat such threats, a task typically assigned to Intrusion Detection Systems (IDS) (Salek et al., 2023). The effectiveness of an IDS largely depends on its ability to accurately identify threats, with a primary goal being the reduction of false positives to enhance reliability (Chougule et al., 2023). Current IDS solutions, however, struggle with the detection of novel attacks and the improvement of operational efficiency. Deep learning techniques stand out for their exceptional efficiency in automated detection, especially renowned for their adeptness at recognizing previously unknown attacks (Sharma & Liu, 2021). These techniques often require data to be formatted into image matrices, a preprocessing step that transforms traditional data into a visual format suitable for DL analysis.

Furthermore, the nature of the IoV, involving vehicles connected over extended periods, challenges conventional models to deliver sustainable results. Deep learning shines in this context, offering high accuracy in identifying malicious activities (Zhang et al., 2024). Nonetheless, the IoV’s reliance on massive data exchanges faces hurdles due to unpredictable wireless channel conditions, where physical obstructions like buildings and bridges may disrupt connectivity. To address these challenges, the development of an adaptable deep learning-based IoV attack detection model is proposed. This model aims to maintain robust detection capabilities in the face of connectivity issues, ensuring the security and reliability of smart car networks (Li et al., 2019). The IoV represents an intricate network enabling communication via the internet among vehicles and infrastructure along the road-side (Al-Hamadi et al., 2020). This advanced field aims to enhance driving experiences, optimize traffic circulation and elevate road safety through the instant transmission of data and sophisticated traffic control systems. Given their proficiency in handling data that follows a sequence, recurrent neural network (RNN) and long short-term memory (LSTM) networks stand out as ideal solutions for analyzing temporal data within IoV systems. These networks excel in detecting irregularities or incorrect sequences in the communication patterns among vehicles, potentially signaling a cyber-security threat (Li et al., 2021). Models such as convolutional neural networks (CNNs) and deep neural networks (DNNs) are derived from deep learning and are instrumental in predicting traffic flow, congestion and driving conditions. These predictions are crucial for optimizing resource allocation, managing traffic and planning routes efficiently (Qin, Xun & Liu, 2024). Furthermore, deep reinforcement learning (DRL) techniques are employed to enhance vehicular networks’ reliability and performance by optimizing resource distribution, including power control, bandwidth management and spectrum allocation. By addressing key challenges and fostering intelligent, efficient and secure vehicle networks, these machine learning algorithms significantly propel forward IoV communications (Satilmiş, Akleylek & Tok, 2024).

The objectives of the proposed attack detection model in IoV network outlined in the article can be summarized as follows: To create a deep learning and heuristic algorithm based attack detection model in IoV communication network, aiming to enhance the security and integrity of vehicular communications.

To implement the EEHLO method for selecting the most relevant features from the collected data. This step is crucial for enhancing the model’s efficiency and accuracy by focusing on the most informative aspects of the data.

To design and develop a Complex-Valued Dilated Recurrent Neural Network (CV-DRNN) for accurately and efficiently detect cyber-attacks within IoV networks. The use of complex-valued neurons is expected to provide superior performance in capturing the intricate patterns and dynamics of cyber-attacks.

To rigorously evaluate the efficiency of the proposed CV-DRNN model using a comprehensive set of metrics. This evaluation compares the model’s effectiveness in attack detection against conventional models to demonstrate its advantages and improvements in IoV security.

In the following sections, provide a detailed exploration of our proposed deep learning framework for detecting attacks within the IoV network. The following set of paragraphs discusses existing research on attack detection models specifically designed for IoV networks, offering insight into the current state of the art. The following section elaborates on the proposed automated attack detection framework from a system perspective, including a thorough description of the dataset used for training and evaluation. As a solution to the proposed system, the next section showcases the implementation of the novel EEHLO method, which plays a critical role in the optimal selection of features for attack identification. Finally, the end of article focuses on the deployment of complex-valued neural networks for the purpose of attack detection in the IoV, highlighting the model’s innovative aspects. The end of article details the deep learning approach adopted in the attack detection model for the IoV network, with an emphasis on the handling and analysis of numerical data.

In 2020, Han, Cheng & Ma (2020) have developed an IDS aimed at improving system security focused on detecting car intrusions using CNN. The system employed an encoder to transform data into an image in real-time, which it then mapped into the complex domains and spun to reconstitute the original features. This section explored the IDS’s capability to conceal features and achieve high-precision acquisition during a vehicle Web assault. Available studies had indicated that attackers could gather attributes from arbitrary angles, complicating the differentiation between genuine and fake features. The proposed method, CNNs-IDS, was found to have a high accuracy rate.

In 2022, Ullah et al. (2022) have proposed a model for IoT detection of network intrusions based on machine learning. However, to effectively and swiftly identify malicious attacks in the IoV in real-time, a more sophisticated method was necessary. As a result of the advancements in IoV technologies, consumers began to pay closer attention to smart cars. Despite the rapid expansion of IoV, numerous privacy and security concerns emerged, posing potential catastrophic risks. The proposed model employed a framework centered around GRU and LSTM to minimize smart vehicle accidents and detect malicious attacks in vehicular networks.

In 2021, Jin et al. (2021) have developed an intrusion detection technique that combined metric learning, outlier identification, and the Synthetic Minority Over-sampling Technique (SMOTE) oversampling technique. By oversampling minority classes using an innovative method, introducing a new feature centered on the discord ratio, actively reducing outliers and rescaling original specimens to improve choice boundary clarity through integrating the detection of outliers and a distance metric studying, the suggested strategy enhanced intrusion detection effectiveness in the following three primary ways.

In 2022, Cheng et al. (2022) have recommended the model made specific use of encoding-detection technology. the encoder portion collected both time and spatial relationships concurrently. Meanwhile, the attention-long memory constructions established important relationships based on past time-series information or significant bytes. Simultaneously, the attention-based convolution network gathered spatial and channel aspects, broadening the field of reception and enhancing the connection between elements. Subsequently, the encoded data was transmitted to the detector, which utilized it to generate powerful spatiotemporal attention characteristics and facilitate the classification of anomalies.

In 2023, Yaqoob et al. (2023) have developed Fog-Assisted IoVs (Fa-IoVs) to identify intrusions. For ensuring smooth Fa-IoVs network interaction, the security risks posed by these networks were treated as abnormalities, which represented a significant challenge. Effective communication in this context meant reduced chances of crucial data loss, delays, communication overhead, etc. “Convolutional Autoencoder Aided Anomaly Detection (CAaDet)”, a deep learning-based approach, was introduced as part of the proposed framework, aiming to address research gaps in the Fa-IoVs network and detect abnormalities. By leveraging convolutional layers and a tailored autoencoder, CAaDet facilitated the identification of anomalies and the extraction of valuable features.

In 2024, Sedjelmaci et al. (2024) have developed to detect hostile IoV devices and edge computers and select suitable IoV devices and edge servers to participate in the detection of attacks and training procedures, thereby strengthening the security of the proposed attack detection architecture, the primary method involved computing a reputation score derived from the actions of these edge servers and IoV devices.

In 2022, Wonjin & Cho (2022) have developed a historical trajectory for identifying intricate attacks. The plan involved using a control center to store behavioral data on all vehicles and roadways. Utilizing this data, a historical trajectory was created, which was then used to identify attacks. The control center would examine the vehicle’s driving track when it was driven irregularly. When a vehicle or the roadway infrastructure underwent an incorrect state change, the control center was able to identify that an attack was being conducted by evaluating the driving process of the vehicle.

In 2024, Yingqing et al. (2024) have developed to address the vulnerabilities of External Vehicular Networks (EVNs) and Intra-Vehicle Networks (IVNs), a lightweight intrusion detection strategy that combined the Hyper-Parameter Optimization (HPO) technique with transfer learning methods, using MobileNetv2 as its foundation.

Problem statement

IoV provides smart assistance to vehicles through the use of sensors, software, and technology that facilitate the wireless exchange of information. However, IoV encounters several challenges such as security, privacy, precise location determination, and diverse quality of service requirements. A major issue is the instability of wireless channel connections, particularly during large data transmissions. As a solution, innovative deep learning-based attack detection methods have been developed for IoV. The challenges and features of existing attack detection models are explained in Table 1. The complex-valued neural network (CVNN) method (Han, Cheng & Ma, 2020) has the ability to accurately identify the known and unknown attack simultaneously. Yet, it lacks the ability to obtain global context information and require long detection time. Long short term memory (LSTM) and gated recurrent unit (GRU) (Ullah et al., 2022) reduce the response and training time. But, slow convergence rate and low learning efficiency. Synthetic Minority Oversampling Technique (SMOTE) (Jin et al., 2021) reduces the risk of overfitting, which often occurs during random oversampling. Even though, false negative error is occurred that leads to false alert. The simulated treatment comparison (STC) (Cheng et al., 2022) method obtains fewer false-alarm rates by maintaining the efficiency of the system. But, this method need full time monitoring, hence it is expensive. Fa-IoVs (Yaqoob et al., 2023) has extensive computing capability. Still, security threats and privacy concern are the major issues. Federated learning (Sedjelmaci et al., 2024) allows for models to be tailored to specific contexts or regions, enhancing the detection of localized attack patterns but the process requires frequent communication between vehicles and a central server, which can be challenging in environments with unstable connections. Deep learning (Wonjin & Cho, 2022) can automatically extract and learn features from raw data, which is beneficial in IoVC where data can be highly dimensional and heterogeneous but it requires large amounts of labeled data for training, which can be difficult to obtain in the context of cyber-attacks. Transfer learning (Yingqing et al., 2024) enables quicker model development and deployment, as it leverages pre-trained models that have already learned useful representations but it requires careful selection and tuning of models. Therefore innovative deep learning based attack detection in IoV is developed.

Table 1 Advantages and Limitations of conventional attack detection in IoV using deep learning.

Author [citation]	Methodology	Advantages	Limitation	
Han, Cheng & Ma (2020)	CVNN	This method has the ability to accurately identify the known and unknown attack simultaneously.

	It lacks the ability to obtain global context information and require long detection time.

	
Ullah et al. (2022)	LSTM and GRU	These methods reduce the response and training time.

Attack detection accuracy is significantly improved.

	Slow convergence rate and low learning efficiency.

	
Jin et al. (2021)	SMOTE	This method reduces the risk of overfitting, which often occurs during random oversampling.

	False negative error is occurred that leads to false alert.

	
Cheng et al. (2022)	STC	This method obtains fewer false-alarm rates by maintaining the efficiency of the system.

	This method needs full-time monitoring, hence it is expensive.

	
Yaqoob et al. (2023)	Fa-IoVs	Fog layer has extensive computing capability.

	Security threats and privacy concern are the major issues.

	
Sedjelmaci et al. (2024)	Federated learning	It allows for models to be tailored to specific contexts or regions, enhancing the detection of localized attack patterns.

	The process requires frequent communication between vehicles and a central server, which can be challenging in environments with unstable connections.

	
Wonjin & Cho (2022)	Deep learning	It can automatically extract and learn features from raw data, which is beneficial in IoVC where data can be highly dimensional and heterogeneous.

	It requires large amounts of labeled data for training, which can be difficult to obtain in the context of cyber-attacks.

	
Yingqing et al. (2024)	Transfer learning	It enables quicker model development and deployment, as it leverages pre-trained models that have already learned useful representations.

	It requires careful selection and tuning of models.

	

Automated model of attack detection in Internet of Vehicles: system view and dataset description

The IoV marks a transformative blend of vehicle networks, the IoT and the broader Internet, paving the way for smart communication among diverse networks and vehicles (Ullah et al., 2022). This fusion aims to enhance the driving experience by offering a plethora of services to drivers and passengers alike. At the heart of this ecosystem are vehicles equipped with on-board units (OBUs), leveraging data from various integrated sensors, like radar, GPS and cameras to aid in decision-making processes. These sensors can identify potential collision threats with bicycles or pedestrians and issue warnings to promote safer driving practices. IoV is underpinned by various forms of networking, each playing a unique role within this interconnected architecture. Vehicle-to-Vehicle (V2V) communication enables cars to wirelessly exchange information, keeping each other informed about their positions and movements. This mutual exchange aims to diminish accident rates by alerting drivers or autonomous vehicles of potential hazards, urging preemptive measures. Vehicle-to-Infrastructure (V2I) communication, on the other hand, establishes a two-way link between vehicles and the surrounding infrastructure. This connection provides drivers access to essential road information through Vehicular Edge Computing (VEC) infrastructure like road-side units (RSUs) and base stations (BSs), enhancing the driving experience tailored to their journey. Furthermore, the VEC network assists OBUs by offloading computational tasks from the vehicles, particularly valuable when vehicle resources are constrained. This infrastructure processes the tasks and relays the insights back to the vehicles, aiding in decision-making and operational efficiency. Infrastructure-to-Infrastructure (I2I) communication facilitates the exchange of data and computational tasks between infrastructure elements such as BSs, RSUs, Fog servers, and Cloud servers, working in concert to optimize the overall performance of the IoV ecosystem. In essence, the IoV ecosystem utilizes these communication networks to foster a more interconnected, safe, and efficient transportation landscape. IoV aspires to revolutionize traffic management, navigation and road safety through enabling intelligent interactions between vehicles and their surrounding infrastructure. Figure 1 shows the structural view of IoV’s system model.

Figure 1 Structural view of IoV’s system model.

Images from Flaticon.com. Pedestrain (image: Flaticon.com, License:Free): https://www.flaticon.com/free-icon/pedestrians_8382946?term=pedestrian&page=1&position=1&origin=search&related_id=8382946 RSU (image: Flaticon.com, License:Free): https://www.flaticon.com/free-icon/signal-tower_2184006?term=satellite+tower&page=1&position=18&origin=search&related_id=2184006 Vehicle (image: Flaticon.com, License:Free): https://www.flaticon.com/free-icon/car_3097136?term=vehicle&page=1&position=1&origin=search&related_id=3097136 Database (image: Flaticon.com, License:Free): https://www.flaticon.com/free-icon/server-storage_3160887?term=database&page=1&position=28&origin=search&related_id=3160887 IPC (image: Flaticon.com, License:Free): https://www.flaticon.com/free-icon/business-center_2676631?term=bank+center&page=1&position=11&origin=search&related_id=2676631 Cloud with up and down arrow (image: Flaticon.com, License:Free): https://www.flaticon.com/free-icon/data_8473799?term=cloud&page=1&position=51&origin=search&related_id=8473799 Base station (image: Flaticon.com, License:Free): https://www.flaticon.com/free-icon/space-station_14604266?term=base+station&page=1&position=64&origin=search&related_id=14604266 Cloud (image: Flaticon.com, License:Free): https://www.flaticon.com/free-icon/cloud_3222791?term=cloud&page=1&position=2&origin=search&related_id=3222791.

In the proposed framework, the initial stage involves the collection of relevant data. In this instance, five datasets are employed to gather the information depicted in Table 2.

Table 2 Attack detection in IoV network model’s-dataset details.

Dataset	Dataset names	Description	
1	CICIDS 2017 KNN Dataset (Sharafaldin, Lashkari & Ghorbani, 2018)
https://paperswithcode.com/dataset/cicids2017
Last accessed 10/8/2024	It is presented as “CICIDS 2017 KNN”. The dataset contains vast data, with detailed features extracted from the captured traffic. This includes over 2 million records.	
2	ISCXNSL-KDD Dataset (Tavallaee et al., 2009)
https://web.archive.org/web/20150205070216/http://nsl.cs.unb.ca/NSL-KDD/
Last accessed 5/4/2024	It is defined as the ISCX NSL-KDD dataset 2009. The NSL-KDD dataset includes features extracted from network traffic with labelled records indicating regular traffic or various types of attacks. The attacks are categorized into four main groups.	
3	KDD-CUP 2019 Dataset (Stolfo et al., 1999)
https://doi.org/10.24432/C51C7N
Last accessed 5/4/2024	The dataset contains a mix of normal and malicious network activity, which are categorized into four main categories. It has been extensively used for training and testing IDS models.	
4	Car Hacking Dataset (Kang et al., 2021)
https://ieee-dataport.org/open-access/car-hacking-attack-defense-challenge-2020-dataset
Last accessed 5/4/2024	It is depicted as the Car Hacking dataset. It consists of network traffic data recorded from the vehicle's internal networks, primarily the CAN bus. This data might include normal operations as well as various attack scenarios (e.g., message injection, DoS attacks on the CAN network, etc.).	
5	UNSW_NB15 (Moustafa & Slay, 2019)
https://ieee-dataport.org/documents/unswnb15-dataset
Last accessed 5/4/2024	It is depicted as the UNSW_NB15 dataset. The UNSW-NB15 dataset comprised 49 features, encompassing source and destination IP addresses, source and destination ports, protocol type, and various other flow-based attributes derived from the traffic. These attributes include duration, byte and packet counts, and the state of the connection.	

Consequently, relevant data associated with the model presented as RTx, where x=1,2,⋯,X, the word X encompassing all the information collected through the dataset.

The integration of the IoT with vehicular networks has given rise to the IoV, a cutting-edge concept aimed at enhancing vehicular communication systems. IoV strives to achieve intelligent traffic management, enhance road safety and improve driving experiences. Attack detection methods play a crucial role in safeguarding the IoV ecosystem against potential cyber threats by continuously monitoring network traffic and vehicle behavior. These models facilitate real-time observation of IoV networks, enabling the swift detection and mitigation of potential threats, which is instrumental in preventing the escalation of security incidents. Effective attack detection models are vital in protecting manufacturers and suppliers from legal and financial repercussions by ensuring compliance with emerging automotive cybersecurity regulations and standards. However, the extensive data monitoring required for efficient attack detection may raise privacy concerns among users, potentially impacting their acceptance of connected vehicle technologies. Moreover, no detection model is infallible and may generate false positives, wrongly identifying harmless activity as malicious and false negatives, failing to recognize actual attacks. These inaccuracies can compromise the models’ effectiveness and efficiency. By implementing the recommended model to minimize these errors, the system’s overall efficiency can be significantly improved and also enhancing the security and reliability of the IoV ecosystem.

Figure 2 shows the brief structural view of deep learning based attack detection in IoV network. This research focuses on the development of a cutting-edge, deep learning-based model for detecting attacks within vehicular networks. By accurately identifying and mitigating potential cyber threats and attacks, this model seeks to safeguard the complex and increasingly vulnerable landscape of vehicular networks. A key component of our approach involves the implementation of the EEHLO method. This method is employed to optimally select the most relevant features from the vast amounts of collected data. This critical step is designed to boost the model’s efficiency and accuracy by homing in on the data that provides the most significant insights into potential security breaches. In response to the unique challenges presented by the IoV networks, propose the design and development of a CV-DRNN. The adoption of complex-valued neurons in the CV-DRNN model is poised to offer unparalleled performance advantages in capturing the subtle and complex patterns characteristic of cyber-attacks within these networks. To validate the efficacy and superiority of our proposed CV-DRNN model, conduct an exhaustive evaluation employing a comprehensive suite of metrics. This thorough assessment will not only benchmark the CV-DRNN model’s attack detection capabilities against those of conventional models but will also highlight its potential contributions to enhancing IoV network security.

Figure 2 Brief structural view of deep learning based attack detection in IoV network.

Adaptive concept of EEHLO

The EEHLO technique utilizes the HLBO algorithm to fine-tune parameters, achieving superior optimization outcomes. HLBO excels at striking a balance between exploration, which involves venturing into new areas of the solution landscape and exploitation, which focuses on enhancing solutions within known territories. The hybrid aspect encourages exploration, while the structured hierarchy facilitates exploitation. This versatility allows HLBO to adapt effectively across diverse challenges by incorporating a variety of optimization approaches. Nonetheless, the complexity inherent in HLBO’s hybrid design might make it more challenging to implement and understand compared to simpler optimization methods. The performance of HLBO is influenced by its features, including the optimization techniques it integrates, necessitating precise calibration that demands expertise and may be time-consuming. To mitigate this limitation, the proposed method aims to reduce errors through strategic application.

Novelty: In the proposed EEHLO approach, after completing rounds of exploration and exploitation, it is advantageous to revisit the exploitation phase once more. This additional exploitation phase allows for further refinement of solutions within familiar regions of the solution space. By doing so, EEHLO can capitalize on the insights gained during exploration and employ them to fine-tune and optimize solutions even more effectively. The computational framework of the recommended EEHLO is outlined as follows:

Algorithm 1 Recommended EEHLO.

Begin HLBO	
Optimization issue data is initialized	
Evaluate objective function	
While (a<Amax)do	
         Phase 1: Exploration	
              To assess the effectiveness of each member in the sample in offering a potential response, use Eq. (4).	
              The hybrid leader for each member of the sample is generated according to Eq. (5).	
              Update the ath member using Eq.(6)	
     Else	
         Phase 2: Exploitation (2 times)	
              The local search process, aimed at enhancing the algorithm’s exploitation capabilities, is modeled using Eq. (7).	
              Update the ath member using Eq.(8)	
     End if	
End while	
Obtain the better candidate solution	
End	

In population-based methods, each individual in the population acts as a seeker within the problem-solving domain, representing a potential solution. The collective capability of the group to propose enhancements is augmented through the algorithmic processes and the exchange of information among its members. However, the method’s exploration of the global solution space can be hindered by the population update mechanism if it depends too heavily on certain members. This situation can lead the algorithm to prematurely converge on a local optimum, thereby obstructing its ability to discover the most optimal solution within the entirety of the search space.

Like other population-based approaches, the HLBO (Dehghani & Trojovsky, 2022) population can be accurately represented by a matrix, in line with Eq. (1).

(1) H=[h1⋮ha⋮hm]M×n=[h1a⋯h1b⋯h1n⋮⋱⋮⋰⋮ha1⋯hab⋯h1n⋮⋰⋮⋱⋮…hM1⋯hMb⋯hMn]M×n.

In this context, the term h refers to the dimensions of the HLBO community, where H represents the total number of variables involved in the problem, ha is the collective group of solutions, hab is the rival solution’ indicates the competing alternative.

Given the constraints outlined in Eq. (2), the starting point for each member of the population is randomly assigned h, taking into account the various limitations.

(2) ha,b=tyb+S.(φb−δb),b=1,2,...n.

Here, the term S specifies the dimensions of the HLBO, where .n signifies the amount of problem. Given the constraints outlined in Eq. (2), the initial position of every population is randomly determined.

The population h denoted by Eq. (3), forms the basis for assessing the objective function of the problem relative to each candidate solution.

(3) L=[L1⋮La⋮LM]M×1=[L(h1)⋮L(ha)⋮L(hM)]M×1.

The variable L represents the objective function; the term La is a measure of the performance of the ath solution of candidate in achieving the goal.

Exploration phase: To determine the extent to which each aforementioned component contributes to shaping the hybrid leader, it’s necessary to evaluate how effectively these contributions support the operation of the goal. To assess the effectiveness of each member in the sample in offering a potential response, use Eq. (4).

(4) Ta=La−Lw∑(La−Lw)a∈(1,2,.....N).

Once the participation rates have been determined, the hybrid leader for each member of the sample is generated according to Eq. (5).

(5) WEa=QWa.Ha+HVb.Hb+WEa.Hf.

The corresponding component will move to the new position if the value of the objective function exceeds that of its previous state; otherwise, it will stay in its original position. This requirement for an update is modeled by Eq. (6).

(6) Ha,b={canew,p1L1new,p1<Licaelse.

Here, the phrase canew,p1 pertains to the new position and function L1new,p1 is the population in the ath iteration, as determined by the objectives established by the hybrid leader.

Exploitation phase (two times): In the HLBO framework, individuals within the algorithm’s ecosystem have the ability to conduct targeted searches in their immediate vicinity for superior solutions, a process known as exploitation. This is facilitated by assigning each member of the population a neighborhood space where they can engage in local search activities to identify positions that yield higher objective function values.

The local search process, aimed at enhancing the algorithm’s exploitation capabilities, is modeled using Eq. (7). Should the position identified through this process result in an improved objective function value, as indicated by Eq. (8), it is deemed suitable.

(7) canew,p1=ca,b+(1−2s)P.(1−1J)

(8) Ha,b={canew,p2L1new,p2<Licaelse.

Here, the phrase canew,p2 pertains to the new position and function L1new,p2 is the population in the ath iteration, as determined by the objectives established by the hybrid leader. Figure 3 shows the flowchart for suggested EEHLO approach.

Figure 3 Flowchart of suggested EEHLO approach.

EEHLO-based optimal features

In the context of the IoV, the phase that involves gathered data RTx as input significantly focuses on enhancing the performance of attack detection systems through optimal feature selection. Optimal feature selection is critical in boosting the efficacy of attack detection systems within the IoV. It is essential to identify and select features that are most indicative of malicious activities. This entails analyzing the data to effectively identify characteristics that are key to distinguishing between regular operations and potential security threats. Without optimal feature selection for attack detection in IoV, several challenges can negatively impact the detection models’ performance and efficiency. For instance, a model incorporating excessive features might overly fit the training data, mistaking random fluctuations for significant patterns. This phenomenon, known as overfitting, hinders the model’s capability to generalize its findings to new, unseen data, thereby diminishing its performance on novel inputs. To address these challenges, optimizing specific parameters such as the number of features extracted from raw data using the recommended EEHLO approach can improve the model’s relief score. The objective function of the proposed model can be formulated as depicted in Eq. (9).

(9) UY1=arg⁡max{Fea}⁡(Rf).

The parameter Fea represents the optimally selected features following the earlier computation, with its range spanning from one to the whole number of features extracted from the raw data. This parameter Rf value serves as a measure for evaluating the relief score. The mathematical expression for the relief score is presented in Eq. (10).

Relief score: This metric assesses the significance of each feature in distinguishing between categories within a dataset. It gauges a feature’s effectiveness in differentiating instances of a specific class from instances of other classes.

(10) ∑Rf=ti−(li−ERi)+(li−YUi).

The term ERi denoted the same class and YUi specifies the various classes.

Attack detection in IoV using complex valued-assisted neural network Dilated RNN

The DRNN (Li et al., 2020) architecture operates within a recurrent framework, similar to how an expanded CNN functions. This model provides a straightforward yet effective solution, tackling multiple challenges simultaneously. It stands out for its incorporation of multi-resolution dilated recurrent connections that implement skipping, characterized by a multi-layered cell-independent structure. The key contributions of this research include:

Introducing an innovative dilated recurrent skip connection feature, addresses gradient issues and enhances the model’s ability to capture long-term temporal dependencies. This new design simplifies the model compared to traditional recurrent mechanisms and significantly boosts computational efficiency.

The construction of a DRNN incorporates various layers of dilated recurrent units with hierarchical expansions, enabling the model to train temporal relationships across different scales and depths more effectively. The introduction of a novel metric, the mean recurrent length, designed to assess the detection capability. This metric facilitates the comparison of efficiency between the enhanced models and previously developed recurrent skip connections. Additionally, the research validates the effectiveness of utilizing an exponentially increasing dilation rate across the network, proving it to be an optimal approach for the proposed architecture.

A depiction of a cell a at a specific layer and time point b is provided Qba. The dilated skip connection formulation details are provided in Eq. (11).

(11) Qba=E(Aba,Qb−ca).

This can be likened to the conventional skip connection, as demonstrated by the equation presented in Eq. (12).

(12) Qba=E(Aba,Qb−ca,Qb−c(y)a).

The skip duration or layer dilation is denoted by a; the input at layer l at that time a is represented by Qb−ca. The DRNN architecture incorporates dilated recurrent layers, as outlined in Eq. (13), to encapsulate complex data dependencies.

(13) Qba=Aba(p−1),b=1,...B.

In this scenario, represents the initial dilation rate Aba(p−1). The final layer incorporates a 1-by-1 convolution layer to compensate for the longer connections that are not captured. Figure 4 shows the architecture view of DRNN followed by the illustration of complex valued dynamic recurrent neural network as proposed in Figure 5.

Figure 4 Architecture view of DRNN.

Figure 5 Proposed view of CV-DRNN for detection.

Suggestive detection model as CV-DRNN

The CV-DRNN represents a cutting-edge approach in neural network design, particularly aimed at boosting cyber-attack detection within the IoV domain. The optimal selected feature Fea is the input to CV-DRNN; this model evolves from the foundational concepts of DRNNs but innovates by incorporating complex numbers into its computations. Traditional DRNNs have been at the forefront of efforts to safeguard this ecosystem by analyzing the extensive data streams from vehicle sensors and communication systems. Despite their capabilities, conventional DRNNs often grapple with the intricate and dynamic nature of IoV data. This data complexity, characterized by its variability over time and across different vehicular systems, poses significant challenges in accurately modeling and predicting cyber-attack patterns. In response to these limitations, an innovative approach leveraging CV-DRNNs has been proposed. The CV-DRNN approach is designed to improve the detection and prediction of cyber threats within the IoV network.

Results and discussion

Simulation setup

Python is a comprehensive environment for numerical computing, played a pivotal role in developing and to assess a proposed attack detection scheme within IoV networks. Additionally, the number of population, chromosomal length and the maximum iteration of the recommended model was 10,5,50. The efficacy of this attack detection approach was benchmarked against other renowned methods, with the findings detailed in subsequent sections. The comparison of results encompassed with both classical optimization techniques, such as the Dingo Optimizer (DO) (Bairwa, Joshi & Singh, 2021), Eurasian Oystercatcher Optimizer (EOO) (Idan, 2023), Tomtit Flock Metaheuristic Optimization Algorithm (TFMOA) (Panteleev & Kolessa, 2022) and HLBO (Dehghani & Trojovsky, 2022), and more contemporary approaches like Random Forest Network (RAN) (Chang, Li & Yang, 2017), LSTM (Ergen & Kozat, 2020), 1DCNN (Wang et al., 2020), recurrent neural network (RNN) (Ullah & Mahmoud, 2022).

Performance evaluation

The implemented attack detection in IoV network uses various metrics and it is given below

Accuracy: Aa=kk+yykk+yy+ss+ll

Specificity: SPE=kkkk+ss

Positive Likelihood Ratio (PLHR): PLHR=kkll

Bookmaker Informedness (BM): BM=kk+yy−1

Fowlkes–Mallows index (FM): FM=kk∗yy

Here, the “true positive and true negative” variables are noted as kkandyy. The “false negative and false positive” variables are noted as ssandll.

Performance analysis on suggested model over different dataset

Figures 6–10 illustrate the performance of the proposed CV-DRNN-based attack detection in IoV networks using datasets 1 (Sharafaldin, Lashkari & Ghorbani, 2018), 2 (Tavallaee et al., 2009), 3 (Stolfo et al., 1999), 4 (Kang et al., 2021), and 5 (Moustafa & Slay, 2019), in comparison with classical classifiers. These figures demonstrate that the suggested network significantly surpasses standard classification methods. Accuracy analysis of the CV-DRNN-based classifier on dataset one revealed notable improvements over classical classifiers. At an epoch value of 400, the CV-DRNN model achieved superior performance, surpassing the RAN by 11.23%, the 1DCNN by 12.3%, the LSTM by 12.3%, and the RNN by 12.8%. This evidence confirms that the CV-DRNN model excels over conventional methods in attack detection for dataset 1. According to the findings, the CV-DRNN model markedly outperformed traditional models in efficiency rate, thus, underscoring the substantial performance advantage of the recommended CV-DRNN-based attack detection in IoV networks over traditional models.

Figure 6 Investigation on developed deep learning based attack detection in IoV network compared over conventional classifier in dataset 1.

(A) Accuracy, (B) bookmaker informedness (BM), (C) Fowlkes–Mallows index (FM), (C) markedness (MK), (D) positive likelihood ratio (PLHR) and (E) specificity.

Figure 7 Investigation on developed deep learning based attack detection in IoV network compared over conventional classifier in dataset 2.

(A) Accuracy, (B) bookmaker informedness (BM), (C) Fowlkes–Mallows index (FM), (C) markedness (MK), (D) positive likelihood ratio (PLHR) and (E) specificity.

Figure 8 Investigation on developed deep learning based attack detection in IoV network compared over conventional classifier in dataset 3.

(A) Accuracy, (B) bookmaker informedness (BM), (C) Fowlkes–Mallows index (FM), (C) markedness (MK), (D) positive likelihood ratio (PLHR) and (E) specificity

Figure 9 Investigation on developed deep learning based attack detection in IoV network compared over conventional classifier in dataset 4.

(A) Accuracy, (B) bookmaker informedness (BM), (C) Fowlkes–Mallows index (FM), (C) markedness (MK), (D) positive likelihood ratio (PLHR) and (E) specificity.

Figure 10 Investigation on developed deep learning based attack detection in IoV network compared over conventional classifier in dataset 5.

(A) Accuracy, (B) bookmaker informedness (BM), (C) Fowlkes–Mallows index (FM), (C) markedness (MK), (D) positive likelihood ratio (PLHR) and (E) specificity.

Evaluation of real and complex valued neural network compared over classical classifier in dataset 1 to 5

Table 3 provides a thorough statistical analysis of multiple optimization algorithms which favoured in obtaining the convergence of the methods across multiple datasets 1, 2, 3, 4 and 5. This analysis is crucial in demonstrating the effectiveness of the EEHLO optimization model in feature selection. Notably, the results outlined reveal that CV-DRNNs outperform several established neural network architectures specifically, RAN, 1DCNN, LSTM and RNN. The superior accuracy metrics surpassing other models by 44%, 88.1%, 34.5% and 78%, respectively, this emphasize the significant enhancements these complex-valued models offer in handling data complexities, potentially transforming approaches in fields requiring precise predictive capabilities and data analysis. This comparative study not only highlights the advanced accuracy of CV-DRNNs but also marks a pivotal advancement in neural network research, promoting further exploration and adoption of complex-valued architectures in broader applications.

Table 3 Dataset statistical report.

Dataset1	
TERMS	DO	EOO	TFMOA	COA	EEHLO	
Worst	8.027	9.067	10.898	4.613	8.470	
Best	4.421	4.309	4.231	4.563	4.068	
Mean	4.712	5.025	4.691	4.585	4.294	
Median	4.421	4.998	4.231	4.563	4.068	
Std	0.731	0.678	1.320	0.025	0.854	
Dataset2	
Worst	10.279	7.839	4.912	4.497	7.230	
Best	4.648	4.228	4.260	4.280	4.050	
Mean	5.002	4.600	4.695	4.293	4.174	
Median	4.648	4.359	4.719	4.280	4.050	
Std	0.926	0.710	0.206	0.051	0.488	
Dataset3	
Worst	7.955	7.419	9.508	10.655	5.311	
Best	4.517	4.239	4.228	4.224	4.061	
Mean	4.644	4.591	4.454	4.505	4.124	
Median	4.517	4.412	4.378	4.224	4.061	
Std	0.622	0.717	0.724	1.039	0.243	
Dataset4	
Worst	10.965	7.471	8.211	7.470	6.715	
Best	4.284	4.378	4.340	4.357	4.172	
Mean	5.364	5.091	4.933	4.625	4.320	
Median	4.565	5.265	4.860	4.689	4.212	
Std	1.578	0.691	0.938	0.436	0.464	
Dataset5	
Worst	6.742	7.935	6.341	6.176	8.561	
Best	4.535	4.220	4.281	4.279	4.017	
Mean	5.028	4.860	4.599	4.486	4.474	
Median	4.618	4.697	4.551	4.279	4.017	
Std	0.733	0.825	0.277	0.567	1.144	

Figure 11 shows the reciever operating characteristics for all the five datasets taken up to consideration and Figure 12 shows the convergence of the optimization of EEHLO.

Figure 11 Receiver operating characteristic curve for datasets 1-5.

(A) Dataset 1 (B) Dataset 2 (C) Dataset 3 (D) Dataset 4 (E) Dataset 5.

Figure 12 Convergence on optimisation algorithms.

(A) Dataset 1 (B) Dataset 2 (c) Dataset 3 (D) Dataset 4 (E) Dataset 5.

Conclusion

The deep learning-based attack detection model tailored for vehicular communications and operations marks a significant step forward in bolstering the cyber security landscape of the IoV. Developing a deep learning-based attack detection model utilizing the EEHLO method for feature selection and a CV-DRNN for attack detection represents a significant leap forward in IoV cyber security. This approach demonstrated the efficacy of complex neural networks in tackling the intricate challenges of cyber-attack detection and established a benchmark for future endeavors in fortifying the security of vehicular networks. This model's successful deployment and validation emphasized the indispensable role of cutting-edge deep learning methodologies in fortifying the resilience of vehicular communications and operations against the relentless evolution of cyber threats. With an epoch value set to 500, the CV-DRNN model exhibited superior performance, surpassing the RAN by 78.23%, the 1DCNN by 89.3%, the LSTM by 34.3%, and the RNN by 77.8% in attack detection. This highlights the CV-DRNN model’s superiority over standard models. The efficacy relies heavily on the quality and quantity of information utilized during training. In scenarios lacking diverse and comprehensive attack data, the model’s ability to generalize and accurately detect novel or sophisticated attacks could be compromised. While the proposed attack detection model presents a promising solution to enhance security in the IoV ecosystem, addressing its current limitations and exploring these future research directions could lead to be more robust, efficient, and adaptable cyber security solutions.

Supplemental Information

Supplemental Information 1 Code.

Additional Information and Declarations

Competing Interests

Author Contributions

Data Availability

Sedat Akleylek is an Academic Editor for PeerJ.

Prasanalakshmi Balaji conceived and designed the experiments, analyzed the data, prepared figures and/or tables, authored or reviewed drafts of the article, and approved the final draft.

Korhan Cengiz performed the computation work, authored or reviewed drafts of the article, and approved the final draft.

Sangita Babu analyzed the data, prepared figures and/or tables, and approved the final draft.

Omar Alqahtani performed the experiments, prepared figures and/or tables, authored or reviewed drafts of the article, and approved the final draft.

Sedat Akleylek performed the computation work, authored or reviewed drafts of the article, and approved the final draft.

The following information was supplied regarding data availability:

The CICIDS 2017 KNN Dataset is available at:

https://paperswithcode.com/dataset/cicids2017.

The ISCXNSL-KDD Dataset is available at:

https://web.archive.org/web/20150205070216/http://nsl.cs.unb.ca/NSL-KDD/.

The KDD-CUP 1999 Dataset is available at:

https://doi.org/10.24432/C51C7N.

The Car Hacking Dataset is available at:

https://ieee-dataport.org/open-access/car-hacking-attack-defense-challenge-2020-dataset.

The UNSW_NB15 Dataset is available at:

https://ieee-dataport.org/documents/unswnb15-dataset.

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
