# Peer review of "Metaheuristic optimized complex-valued dilated recurrent neural network for attack detection in internet of vehicular communications"

_PeerJ Computer Science, doi:10.7717/peerj-cs.2366_

## Round 0.1 · original submission · Major Revisions

In the opinions of three reviewers and mine, a major revision can be recommended for this paper.

Reviewer 1 ·

Basic reporting

The manuscript entitled “Metaheuristic optimized complex-valued Dilated Recurrent Neural Network for attack detection in Internet of Vehicular Communications” has been investigated in detail. The paper proposes an innovative attack detection model for the Internet of Vehicles (IoV) using advanced deep learning techniques. While the topic is relevant and timely, addressing critical security issues in vehicular networks, the paper has several weaknesses that need to be addressed. Below is a detailed critique highlighting the key issues and suggestions for improvement.
1) The introduction lacks a clear and specific problem statement. It is essential to explicitly state the problem that the proposed model aims to solve, emphasizing the significance of the issue.
2) The background information on IoV and the importance of attack detection is too general. More detailed context and examples of specific challenges in IoV security would strengthen the motivation.
3) The literature review is superficial and does not adequately cover existing work in the field. A comprehensive review of state-of-the-art attack detection methods in IoV, highlighting their strengths and limitations, is necessary.

Experimental design

4) The paper should include a detailed comparison of the proposed method with existing techniques, clearly identifying the novel contributions of this work.
5) The methodology section lacks detail. The description of the Enhanced Exploitation in Hybrid Leader-based Optimization (EEHLO) method and Complex-Valued Dilated Recurrent Neural Network (CV-DRNN) should be more detailed, including the underlying principles, architecture, and parameters.
6) The feature extraction process needs to be explained in more detail. The paper should specify what features are extracted, why they are chosen, and how they contribute to the detection model.
7) The authors should clearly emphasize the contribution of the study. Please note that the up-to-date of references will contribute to the up-to-date of your manuscript. The studies named- “Overcoming nonlinear dynamics in diabetic retinopathy classification: a robust AI-based model with chaotic swarm intelligence optimization and recurrent long short-term memory; Artificial intelligence-based evaluation of the factors affecting the sales of an iron and steel company”- can be used to explain the methodology and optimization process in the study or to indicate the contribution in the “Introduction” section.

Validity of the findings

8) The paper should provide comprehensive information on the training and validation process, including the dataset used, training duration, hyperparameter tuning, and any pre-processing steps.
9) The paper should discuss the practical implications of the findings, including potential challenges in real-world deployment and how they might be addressed.

Reviewer 2 ·

Basic reporting

The paper presents a novel approach to attack detection in Internet of Vehicular Communications using a complex-valued dilated recurrent neural network optimized with metaheuristics. Overall, the methodology and results are promising. The authors provide good context and background on IoV security challenges and existing approaches. The paper is generally well-structured with appropriate use of figures and tables to illustrate key concepts and results. However, there are a few areas that could be improved with minor revisions. The English writing is mostly clear but would benefit from some polishing to enhance readability and professionalism throughout. Some technical terms and acronyms could be more clearly defined when first introduced. The literature review is comprehensive, but tighter integration of cited works into the motivation for the proposed approach would strengthen the paper. While experimental results are presented in detail, more rigorous statistical analysis and comparison to baselines would bolster the conclusions. Additionally, sharing the raw data and code used would increase reproducibility. With these minor enhancements, this paper could make a solid contribution to the field of IoV security.

Experimental design

no comment

Validity of the findings

no comment

Additional comments

n/a

Reviewer 3 ·

Basic reporting

All comments have been added in detail to the last section.

Experimental design

All comments have been added in detail to the last section.

Validity of the findings

All comments have been added in detail to the last section.

Additional comments

Review Report for PeerJ Computer Science
(Metaheuristic optimized complex-valued Dilated Recurrent Neural Network for attack detection in Internet of Vehicular Communications)

1. Within the scope of the study, an attack detection model specific to the study has been developed in order to create more secure transportation networks via the internet of vehicles using deep learning techniques.

2. In the introduction section, the internet of vehicles, its importance in the literature, the contributions of the attack detection model proposed in the study to the literature and the literature review are mentioned at a certain level. In this section, it is recommended to mention the methodologies, preprocessing processes and results of the studies specified in Table-1 in more detail by adding them to the table.

3. Both the internet of vehicles system model and the attack detection model developed based on deep learning in the internet of vehicles network are explained by specifying them at the appropriate level. However, in this section, although there are many different optimization techniques that can be used when the literature is examined, it should be stated more clearly why the "enhanced exploitation in hybrid leader-based optimization" technique is preferred and what its superiority and originality are compared to other techniques in the literature.

4. Using more than one dataset for the proposed attack detection model increases and proves the quality of the study and the usability of the model.

5. When the details and structure of the complex-valued dilated recurrent neural network used in the study are examined, it is observed that it is suitable and sufficient for the study.

6. When the types of evaluation metrics obtained for the analysis of the results and the results are examined, it is both sufficiently detailed and reveals the superiority of the proposed model.

As a result, this study has the potential to make a significant contribution to the literature, and it is recommended to examine the sections listed above.

---

## Round 0.2 · accepted · Accept

In the opinions of reviewers and mine, this revised paper is able to accept.

Reviewer 1 ·

Basic reporting

It is acceptable in the present form.

Experimental design

It is acceptable in the present form.

Validity of the findings

It is acceptable in the present form.

Reviewer 3 ·

Basic reporting

All comments have been added in detail to the last section.

Experimental design

All comments have been added in detail to the last section.

Validity of the findings

All comments have been added in detail to the last section.

Additional comments

Thank you for the revision. The changes made are sufficient. I recommend that the paper be accepted.